



# Seasonal shifts in export of DOC and nutrients from burned and unburned peatland-rich catchments, Northwest Territories, Canada

Katheryn Burd[1], Suzanne E Tank[2], Nicole Dion[3], William L Quinton[4], Christopher Spence[5], Andrew J Tanentzap[6], David Olefeldt[1]

[1]Department of Renewable Resources, University of Alberta, Edmonton, T6G 2R3, Canada
[2]Department of Biological Sciences, University of Alberta, Edmonton, T6G 2E9, Canada
[3]Water Resources Department, Government of Northwest Territories, Yellowknife, X1A 2L9, Canada
[4]Centre for Cold Regions and Water Science, Wilfred Laurier University, Waterloo, N2L 3C5, Canada
[5]National Hydrology Research Centre, Environment and Climate Change Canada, Saskatoon, S7N 3H5, Canada
[6]Ecosystems and Global Change Group, Department of Plant Sciences, University of Cambridge, Cambridge, CB2 3EA United Kingdom

*Correspondence to*: olefeldt@ualberta.ca

**Abstract.** Boreal peatlands are major catchment sources of dissolved organic carbon (DOC) and nutrients, and thus strongly regulate the landscape carbon balance, aquatic food webs, and downstream water quality. Climate change is likely to
influence catchment solute yield directly through climatic controls on runoff generation, but also indirectly through altered disturbance regimes. In this study we monitored water chemistry from early spring until fall at the outlets of a 321 km$^2$ catchment that burned 3 years prior to the study and a 134 km$^2$ undisturbed catchment. Both catchments were located in the discontinuous permafrost zone of boreal western Canada and had ~60% peatland cover. The two catchments had strong similarities in the timing of DOC and nutrient yields, but a few differences were consistent with anticipated effects of
wildfire based on peatland porewater analysis. The four week spring period, particularly the rising limb of spring freshet, was crucial for accurate characterization of the seasonal solute yield from both catchments. The spring period was responsible for ~65% of the seasonal DOC and nitrogen, and ~85% of the phosphorous yield. The rising limb of spring freshet was associated with high phosphorous concentrations and DOC of distinctly high aromaticity and molecular weight. Shifts in stream DOC concentrations and aromaticity outside the early spring period were consistent with shifts in relative
stream-flow contribution from precipitation-like water in the spring, to mineral soil groundwater in the summer, with consistent relative contributions from organic soil porewater. Radiocarbon content ($^{14}$C) of DOC at the outlets was modern throughout May to September (fM: 0.99-1.05), but likely reflected a mix of aged DOC, e.g. porewater DOC from permafrost (fM: 0.65-0.85) and non-permafrost peatlands (fM: 0.95-1.00), with modern bomb-influenced DOC, e.g. DOC leached from forest litter (fM: 1.05-1.10). The burned catchment had significantly increased total phosphorous yield, and also had greater
DOC yield during summer which was characterized by a greater contribution from aged DOC. Overall, however, our results suggest that DOC composition and yield from peatland-rich catchments in the discontinuous permafrost region likely is more sensitive to climate change through impacts on runoff generation rather than through altered fire regimes.





## 1 Introduction

Catchment export of terrestrially derived dissolved organic carbon (DOC) and nutrients represent significant losses from terrestrial ecosystems and further regulate functions of downstream aquatic ecosystems, including primary productivity, light conditions, heterotrophic respiration, greenhouse gas emissions, and availability of contaminants such as mercury (Karlsson
et al., 2009; Tranvik et al., 2009; Braaten et al., 2014). Peatlands are important sources of DOC and nutrients in boreal catchments, due to large stores of soil organic matter in peatlands that often are hydrologically well connected to stream networks (Laudon et al., 2011). The discontinuous permafrost zone of boreal western Canada contains some of the largest and most extensive peatlands in the circumboreal region (Tarnocai et al., 2009), and the region is experiencing rapid climate change. In order to anticipate impacts of climate change on catchment DOC and nutrient export patterns in boreal western
Canada, it is likely required to take into account both regional characteristics that influence catchment hydrology, e.g. the relatively dry climate and the presence of permafrost, as well as the role of disturbances, e.g. an intensified fire regime (Flannigan et al., 2009).

Water chemistry of boreal streams exhibits large variability both in time and space due to differences in hydrological
connectivity of various catchment water sources. The relative catchment coverage of peatlands with thick organic soils, and upland forests with mineral soils, is a first order control on stream water chemistry in boreal regions (Laudon et al., 2011). Peatlands often act as major sources of dissolved organic matter due to the direct hydrological connectivity of their large stores of soil organic matter, and can dominate cumulative annual catchment DOC export if peatland coverage exceeds 10% (Ågren et al., 2008). The temporal variability in stream water chemistry is often associated with variability in catchment
runoff generation, where the relationships between chemistry and runoff can be used to infer hydrological processes and indicate the relative contribution from various water sources under different conditions (Godsey et al., 2009; Ågren et al., 2014). For example, during periods of high runoff it has been found that peatland runoff generally is diluted while runoff generation in upland forest riparian zones leads to hydrological connection with the shallow organic soil which causes increased stream concentrations of DOC and nutrients (Carey, 2003; Ledesma et al., 2017; Ågren et al., 2008). As such it is
crucial to monitor stream water chemistry during high runoff periods, including spring freshet in high latitude regions, in order to accurately characterize the cumulative catchment export of DOC and nutrients to downstream ecosystems (Finlay et al., 2006).

The chemical composition of dissolved organic matter in streams can indicate both the contribution of distinct sources to
DOC and nutrient export (Wickland et al., 2007), as well as influence its susceptibility to microbial and photochemical processing or flocculation in downstream aquatic ecosystems (Sulzberger and Durisch-Kaiser, 2009; von Wachenfeldt et al., 2009; Cory and Kaplan, 2012). Easy-to-measure spectrophotometric indicators of bulk DOC aromaticity (e.g. specific UV absorbance at 254 nm [Weishaar et al., 2003]) and average molecular size (e.g. spectral slope between 275 and 295 nm





[Helms et al., 2008; Fichot and Benner, 2012]) have been found to be useful for both differentiating catchment water sources, and to indicate microbial and photochemical reactivity. While DOC derived from organic soils, including peat and riparian soils, generally has higher aromaticity and molecular weight than DOC derived from mineral soil groundwater sources (Kaiser and Kalbitz, 2012), there is still a wide variability depending on degree of soil humification, type of vegetation litter, season, and fire history (Wickland et al., 2007; Hugelius et al., 2012; Olefeldt et al., 2013c; O'Donnell et al., 2016). Different catchment DOC sources may also vary in terms of radiocarbon ($^{14}C$) age (Raymond et al., 2007), which provides another mean to assess contributing sources to catchment DOC export. However, DOC radiocarbon age further indicates the immediacy of links between terrestrial and aquatic biogeochemistry (Campeau et al., 2017), and in permafrost regions the potential downstream mobilization of previously frozen soil organic matter (Spencer et al., 2015). Shifts in stream DOC composition due to climate change, either directly or through impacts of wildfire, may thus be as important for downstream ecosystems as a shift in export magnitude.

Approximately 25% of treed permafrost-affected peat plateaus have burned during the last 30 years in the discontinuous permafrost zone of western boreal Canada (Gibson et al., In review). The fire regime in this region is thus already showing signs of increased occurrence due to climate change, and a 50% increase in fire occurrence is projected by the end of the century (Flannigan et al., 2009). While fire has been found to generally cause increased catchment export of total phosphorous in the boreal biome, impacts on DOC and total dissolved nitrogen include increases, decreases, and no change depending on which boreal region studies have been carried out in (Carignan et al., 2000; Lamontagne et al., 2000; McEachern et al., 2000; Betts and Jones, 2009; Marchand et al., 2009). Wildfire does not cause complete combustion of peatland soils in the discontinuous permafrost zone in boreal western Canada, given the significant depth of the peat deposits. However, wildfire in permafrost affected peat plateaus has been shown to cause significantly deepened seasonally thawed peat layer above the permafrost, i.e. the active layer, (Gibson et al., In review) which suggests that peatland runoff in the years following fire is routed through deeper, older peat layers. As such, catchments in this region may exhibit characteristic response to wildfire with regards to DOC composition.

The objective of this study was to improve our understanding of controls on catchment DOC and nutrient export in a region with extensive permafrost-affected peatland complexes, and to assess potential impacts of recent wildfires. We monitored DOC and nutrient export from two catchments from early spring freshet to late fall, one of which was almost completely burned three years prior to the study, and further collected peatland pore water throughout the study within burned and unburned peatland sections. High frequency records of stream chemistry from spring to fall are uncommon from high latitude catchments, and from the study region in particular, given the practical challenges of working in remote locations. We hypothesized low overall catchment yields of DOC and nutrients when compared to other boreal peatland-rich regions due to the relatively dry climate, and that a majority of yields were associated with spring freshet. We further hypothesized





modest impacts of wildfire on the magnitude of catchment DOC yields, but increased DOC aromaticity and radiocarbon age due to deepened flow paths within burned peat plateaus.

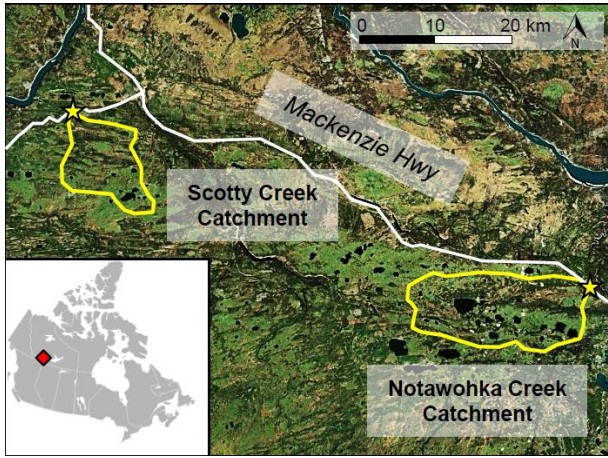

**Figure 1: Scotty Creek and Notawohka Creek catchments in the discontinuous permafrost zone of boreal western Canada. Sampling locations at outlets indicated by yellow stars. Photo source: Zoom Earth MODIS Satellite.**

## 2 Methods

### 2.1 Catchment descriptions

Scotty Creek (SC) and Notawohka Creek (NW) drain catchments located in the discontinuous permafrost zone of Canada's western boreal forest (Fig. 1). The SC outlet at the Liard Highway (61°24 W, 121°26 N) has a 134 km$^2$ catchment that has not been affected by any major fires in the last 60 years. The NW outlet at the Mackenzie Highway (61°08 W, 120°17 N) has a 321 km$^2$ catchment that was >90% burned in 2013 (Northwest Territories Fire Scar Map, 2013). Both catchments are located in the Taiga Plains Mid-Boreal Ecoregion within the Mackenzie Basin (Ecosystem Classification Group, 2007), which is characterized by short summers and long, dry winters (Quinton et al., 2009). Mean annual temperature is -3.5°C and mean annual precipitation is 350 mm, with the majority falling as snow (Meteorological Services of Canada, 2002; Ecosystem Classification Group, 2007; Quinton et al., 2009). The bedrock of the catchments is dominated by sedimentary shale, limestone, and dolomite rocks formed during the Devonian period (Wheeler et al., 1996), while surface geology is dominated by thin to thick tills and glaciolacustrine fine-grained deposits formed during glacial retreat after the last glacial maximum (Aylsworth et al., 2000). Both catchments are flat to gently undulating with < 50 m difference between outflow and maximum elevation (248 - 295 m and 263 - 300 m above sea level for SC and NW, respectively). The ecoregion is co-dominated by mixed-wood forests of trembling aspen (*Populus tremuloides*) and white spruce (*Picea glauca*) in well-drained locations, and extensive lowland peatlands where peat deposits can be up to 8 m thick (Quinton et al., 2009). The peatlands are a mosaic of permafrost peat plateaus, permafrost-free thermokarst bogs and channel fens (Quinton et al., 2009; Wieder





and Vitt, 2010). Peat plateaus are relatively dry and dominated by black spruce (*Picea mariana*), Labrador tea (*Rhododendron groenlandicum*), and a variety of lichen species; thermokarst bogs are mesic in wetness and support *Sphagnum spp* mosses and low shrubs, while channel fens have a persistent water table above the soil surface and vary from being dominated by sedges and other tall graminoids to being dominated by shrubs mostly from the genus *Betula* (Quinton et al., 2009; Wieder and Vitt, 2010). Neither thermokarst bogs nor channel fens carry fire well, thus these ecosystems were largely unaffected by the fire that burned the Notawohka Creek catchment.

We used digital elevation models and satellite image interpretation to delineate the stream networks and the catchment limits. A supervised land cover classification (ArcGIS 2017, ESRI, Redlands, CA, USA) of the SC and NW catchments was carried out using publicly available MODIS satellite imagery taken before the 2013 fire (Zoom earth, https://zoom.earth/, accessed April 2017). Easily distinguishable spectral signatures were utilized to identify four land cover classes: 1) open water, 2) upland mixed-wood forests, 3) channel fens, and 4) peat plateau and thermokarst bog complexes (Fig. 2). Image resolution was too coarse to further separate peat plateaus and thermokarst bogs. The two catchments had similar proportional contributions from different land cover classes; both had ~60% peatland coverage dominated by peat plateau and thermokarst bog complexes (Fig. 2). The NW catchment had more open water, and a greater proportion of catchment runoff passes through a series of lakes.

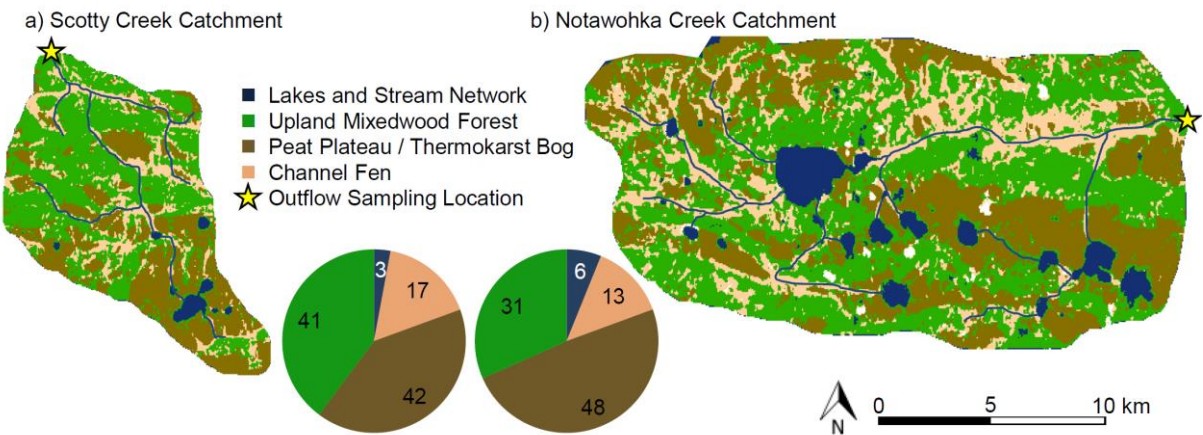

**Figure 2: Land cover distribution and stream networks within a) Scotty Creek and b) Notawohka Creek catchments. Land cover classification was done through supervised classification using maximum likelihood methodology. Pie charts indicate the percent coverage of each land cover type within each catchment.**

## 2.2 Catchment discharge and runoff

Air pressure and pressure at the bottom of the water column were continuously monitored at one-hour intervals with HOBO Pressure Loggers (Onset, Bourne, MA, USA) from April 29th to September 9th, 2016 at the SC and NW catchment outlets. The continuous water stage record was verified by taking stage measurements at stationary staff gauges throughout the study




period. The rating curve for estimating hourly discharge at the NW outlet was based on seven manual measurements of discharge (SonTek Flowtracker Handheld ADV, San Diego, CA, USA). Discharge was estimated using the velocity-area method, where stream velocity at 60% depth was multiplied by cross-sectional areas at 10 locations along a transect spanning the width of the stream (Shaw, 1994). Discharge was measured both during peak spring conditions and summer

base flow conditions, and the resulting rating curve had an $R^2$ of 0.96. We estimated an hourly discharge record for the study period using the rating curve and the hourly record of stream stage. Issues with ice damming in the channel during early spring made the rating curve inappropriate, and instead we estimated hourly discharge through linear interpolation between manual discharge measurements from the start of the study until May 7[th]. Discharge at the SC outlet has been monitored by Water Survey of Canada since 1995 (wateroffice.ec.gc.ca) (Fig. 3), but we also measured stream discharge at the SC outlet

on seven occasions during the study period to ensure consistency with the NW discharge data. Our discharge measurements at SC were well correlated and had minimal offset when compared to the Water Survey of Canada record ($R^2 = 0.99$). Catchment runoff (mm d$^{-1}$) was calculated by dividing the daily discharge rate by catchment area.

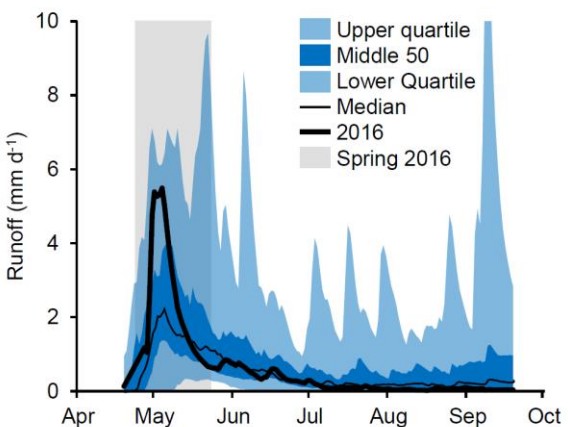

**Figure 3: Comparison of the 2016 hydrograph with the historical record from 1995-2015 at the Scotty Creek outlet. Historical**
**runoff data maintained by Water Survey of Canada.**

**2.3 Monitoring of water chemistry and catchment solute yields**

Water chemistry at SC and NW outlets was monitored using a combination of grab samples for nutrient and DOC analysis, and continuous measurements of electrical conductivity (Ec), temperature, and absorbance over the full UV-vis spectra. Grab samples were collected on sixteen occasions between April 29[th] and September 9[th], 2016, with 8 samples collected in the first

month. Two 60 mL samples were taken at each occasion, which were filtered immediately through 0.7 μm nominal pore size glass fibre filters (Grade GF/F, Whatman) into amber glass bottles. Samples were acidified with 0.6 mL 2 N HCl in order to reduce pH to < 2 and thus prevent further microbial activity during transport to the laboratory. An additional 4 mL filtered, but not acidified, sample was transferred to a 1 cm path-length quartz cuvette and analysed for UV-vis absorbance between 230 and 600 nm using a field portable spectrophotometer (Flame-DA-CUV-UV-VIS light source and Flame-S





Spectrophotometer, Ocean Optics, Dunedin, FL, US). Acidified samples were analysed within eight days of collection for concentrations of DOC and total dissolved nitrogen (TDN) on a TOC-L combustion analyser with a TNM-L module (Shimadzu, Kyoto, Japan), using four injections where the average standard deviation for injections of the same sample was 0.07 mg C $L^{-1}$ and 0.008 mg N $L^{-1}$. Standard concentration solutions and blanks (Milli Q water) were included in each run to

ensure accuracy and to avoid analytical drift. An additional 8 samples were analysed photometrically (690 nm) for total phosphorous (TP) and total dissolved phosphorous (TDP) concentrations through stannous chloride method.

Continuous measurements (4 hour intervals) of decadal absorbance ($cm^{-1}$) between 200 and 700 nm, every 2.5 nm, began on April 29[th] at both catchment outlets using spectro::lyser[TM] (s::can Messtechnik GmbH, Vienna, Austria). The instruments

were adapted for water with high absorbance with narrow 5 mm path-lengths, and were outfitted with automatic brushes that cleaned the lenses 1 minute before every measurement. Unfortunately, the automatic brush on the instrument at the SC outlet was damaged a few days after being deployed due to upstream ice break-up. The instrument was restored on May 19[th], but lens fouling became an issue and data after June 21[st] was discarded. The instrument at the NW outlet functioned properly until August 17[th], when there was a battery malfunction which prevented further data collection. Quality of the spectro::lyser

data was checked by comparison with UV-vis data from grab samples analysed on the field portable UV-vis instrument. Agreement between the spectro::lyser and grab sample data required correction of the spectro::lyser data for turbidity. Following common practice, turbidity was assumed to cause light scattering independent of wavelength, while absorbance of dissolved organic matter was assumed to have negligible absorbance at wavelengths > 550 nm. Hence, the average absorbance at wavelengths between 550 and 700 was subtracted from the full spectro::lyser UV-vis absorbance spectra

(Green and Blough, 1994). Correlation between corrected absorbance from the spectro:lysers and absorbance of grab samples at 254 nm had an $R^2$ of 0.95 with < 5% offset. Continuous measurements (1 hour intervals) of temperature and electrical conductivity were logged at both outlets throughout the study period using HOBO Freshwater conductivity loggers (Onset, Boruner, MA, USA). The continuous Ec record was checked by comparison against manual Ec measurements taken during each sampling occasion (YSI Professional Plus Multiparameter Water Quality Instrument, Yellow Springs, Ohio,

USA).

We assessed relative DOC aromaticity and average molecular weight using the specific UV absorbance at 254 nm (SUVA, L mg $C^{-1}$ $m^{-1}$) and spectral slope between 275 and 295 nm ($S_{275-295}$, $nm^{-1}$), respectively. The SUVA was calculated for grab samples by dividing decadal absorbance at 254 nm ($cm^{-1}$) with DOC concentration (mg C $L^{-1}$), multiplied by 100 (Weishaar

et al., 2003). A higher SUVA indicates higher DOC aromaticity. $S_{275-295}$ was estimated on data from the spectro::lyser using a linear fit of the log-linearized spectra between 275 and 295 nm (Fichot and Benner, 2012). $S_{275-295}$ has been inversely linked to average molecular weight (Obernosterer and Benner, 2004; Helms et al., 2008).



Daily catchment yield of DOC, TDN, TDP, and TP (mass $m^{-2}$ $d^{-1}$) was estimated by multiplying catchment runoff (mm $d^{-1}$) with solute concentrations that were linearly interpolated for periods between each grab sample. Cumulative solute yield was calculated for the rising limb of spring freshet, the four week spring period, and for the entire study period. Catchment yield of DOC for SC was also estimated for the period 1995 to 2015 using the historical runoff record and the relationship

between runoff and DOC concentration observed in 2016 ($R^2 = 0.71$, $p < 0.005$). These estimated allowed comparison of the magnitude and timing of DOC yield in 2016 with likely patterns in previous years.

**2.4 End-member mixing analysis to determine streamflow contributions**

A hydrograph separation was carried out for both SC and NW streamflow using three potential water sources as end-members; organic soil porewater, mineral soil groundwater, and precipitation-like water. We chose to use $A_{254}$ and Ec as the

two tracers for the hydrograph separation. We assumed conservative mixing for both $A_{254}$ and Ec, which implies an overall negligible influence of processes like photodegradation and solute precipitation. High frequency data were available for both $A_{254}$ and Ec for periods when the spectro::lysers were functioning, while $A_{254}$ from grab samples were used when spectro::lyser data were missing. The organic soil porewater end-member was characterized based on porewater that was collected from different peatland ecosystems in the region throughout the 2016 summer, see below. While the organic soil

porewater end-member was characterized using peatland porewater ($A_{254}$: 3.0 ±0.5 $cm^{-1}$ [±95%CI], Ec: 55 ± 10 µS $cm^{-1}$), it was assumed to also be representative of runoff generated in riparian zones of upland forests during very wet periods when runoff is routed through shallow organic soils (Seibert et al., 2009). Such shallow riparian organic soil porewater has been found to have similar characteristics as peatland porewater in other boreal regions, with $A_{254}$ > 1.7 $cm^{-1}$ and Ec < 110 µS $cm^{-1}$ (Jantze et al., 2015; Kothawala et al., 2015). Mineral groundwater and precipitation-like water sources were characterized

using published values from studies in the same region (Hayashi et al., 2004; Fraser et al., 2001). The precipitation-like water source was thus assumed to have $A_{254}$ of 0.05 ±0.03 $cm^{-1}$ and Ec of 15 ±10 µS $cm^{-1}$ (±95% CI) while mineral soil groundwater was assumed to have $A_{254}$ of 0.25 ±0.10 $cm^{-1}$ and Ec of 325 ±175 µS $cm^{-1}$. The flow separation of three end-members and two tracers was solved using Eq. 1-3 (Christopherson and Hooper, 1992):

$$R_T = R_O + R_M + R_P \tag{1}$$
$$C_T^{A254} \times R_T = C_O^{A254} \times R_O + C_M^{A254} \times R_M + C_P^{A254} \times R_P \tag{2}$$
$$C_T^{EC} \times R_T = C_O^{EC} \times R_O + C_M^{EC} \times R_M + C_P^{EC} \times R_P \tag{3}$$

where $R_T$ is the measured catchment runoff, $R_O$, $R_M$, and $R_P$ are the fractional runoff contribution from organic soil

porewater, mineral soil groundwater, and precipitation-like end-members, respectively. $C_T^{A254}$ and $C_T^{EC}$ are the measured $A_{254}$ and Ec at catchment outlets, and $C_S$, $C_G$, and $C_P$ denote $A_{254}$ and EC for each end-member. Uncertainties of end-member contributions to streamflow were assessed by solving Eq.1-3 using combinations of end-member $A_{254}$ and Ec 95% CI that yielded minimum and maximum streamflow contributions.





## 2.5 Peatland porewater sampling

Peatland porewater was collected on three occasions during the study period (early June, late July, and early September) in order to assess impacts of wildfire on porewater chemistry, and to characterize the organic soil porewater end-member. Porewater was collected from a large peatland complex which was accessible by foot from the road (61.19˚N, 120.08˚W).

Parts of the peatland complex had been affected by the 2013 Notawohka fire. Porewater was collected from non-permafrost thermokarst bogs and permafrost peat plateaus – in both burned and unburned sections. At each site, we dug three pits down to the water table (~10 cm in the thermokarst bog and, ~50 cm on the peat plateaus) and let the water fill in and particulates to settle before collecting water samples in the same way as stream water samples, described above. Water was collected near the water table position as this porewater is most likely to be laterally mobile and thus most likely to contribute to

downstream streamflow. Ec was measured directly in the dug pits using a hand-held Ec meter after water collection. Peatland porewater was analysed for UV-vis absorbance and concentrations of DOC, TDN, and TDP, as described above.

## 2.6 Radiocarbon dating of DOC and forest-floor litter

We sampled and analysed radiocarbon content of DOC from the catchment outlets and peatland porewater, and of forest-floor litter in 2017, i.e. during the year following the main study period. The SC hydrograph in 2017 was similar to 2016,

with no major stormflow during the summer. We collected stream water from both the SC and NW outlets on three occasions in 2017; May 9[th], July 7[th], and September 11[th]. On each occasion, we collected three replicate 2 L samples which were filtered in the field through pre-baked 0.7 μm filters (475˚C for 4 h) using a pre-baked glass filtration assembly (500˚C for 4 h) Peatland porewater samples for DOC radiocarbon analysis were collected once, in early September, at the same sites as porewater had been collected in 2016. We dug pits down to the water table, and then inserted ten MacroRhizones with a

0.15 μm pore size (Rhizosphere Research Products, Wagening, the Netherlands) into the peat at the level of the water table in each pit for porewater extraction. Pre-combusted bottles (0.5 L) were filled at each pit and then stored dark and cool. We also collected triplicate mixed, representative, forest-floor litter samples from upland mixedwood forest, thermokarst bogs, burned peat plateaus, and unburned peat plateaus. Approximately 20 g of litter was collected in each sample. Samples were dried and ground up in the lab. The radiocarbon signature of DOC was measured following extraction and purification at the

A.E. Lalonde AMS Laboratory, Ottawa, Canada, using a 3MV tandem accelerator mass spectrometer (High Voltage Engineering) following established methodologies (Lang et al., 2016; Palstra and Meijer, 2014; Zhou et al., 2015, Crann et al., 2017), and is reported with an error estimate of 1 sigma. The fraction modern carbon (fM) relative to 1950 was calculated according to Reimer et al. (2004) from the ratio of the sample $^{14}C/^{12}C$ to $^{14}C/^{12}C$ of an oxalic acid II standard measured in the same data block. Both $^{14}C/^{12}C$ ratios were background-corrected using the AMS-measured $^{13}C/^{12}C$ ratio.



## 3 Results

### 3.1 Catchment runoff

We estimated 84 and 92 mm in total runoff between April 29th and September 9th, 2016, from the SC and NW catchments, respectively (Table 1). This indicated no detectable difference in total runoff from the two catchments, given the uncertainty

of the rating curve at the NW outlet at high discharge. The seasonal hydrograph was dominated by freshet, and we defined the period April 29th until May 26th as the spring period, since this marked the end of freshet and any subsequent rise in runoff occurred in response to rain (Fig 4). Peak spring runoff from the SC catchment occurred on May 5th (4.8 mm d$^{-1}$), and on May 8th from the larger NW catchment (4.7 mm d$^{-1}$). Total runoff during the spring period was 62 and 61 mm from the SC and NW catchments, respectively, representing 75 and 67% of the total study period runoff. Runoff during the summer

period was characterized mainly by base flow conditions, interrupted by a few minor storm events (Fig. 4). The NW catchment sustained greater runoff during the summer period, with 21 and 31 mm runoff from the SC and NW catchments, respectively.

Precipitation and air temperature data were available from the Fort Simpson meteorological station, located 50 km northwest

of the Scotty Creek catchment outlet (climate.weather.gc.ca). In 2016, mean air temperature between May and September was 16.0˚C, compared to the 1980-2010 average of 14.0˚C. Precipitation as snow in the winter preceding the study, from October 2015 until May 2016, was 153 mm compared to the long-term average of 152 mm, while precipitation as rain from May until September during the study in 2016 was 169 mm compared to the long-term average of 203 mm. Reflecting the normal snow-accumulation prior to freshet and the drier and warmer climatic conditions throughout May to September, we

found that spring runoff from the SC catchment in 2016 was similar to the 1995-2015 average (62 vs 61 mm), while summer runoff in 2016 was much lower than the long-term average (21 vs 65 mm). The period from start of spring freshet until the end of our monitoring (September 9th) has over the long-term record accounted for 86% of the annual runoff at the SC outlet.





**Table 1. Cumulative runoff and solute yield from Scotty Creek and Notawohka Creek catchments during the 2016 study period.**

|  | Scotty Creek Catchment | Notawohka Creek Catchment |
|---|---|---|
| Cumulative runoff | 84 mm | 92 mm |
| Spring runoff[1] | 62 mm (75%) | 61 mm (67%) |
| Summer runoff[2] | 21 mm (25%) | 31 mm (33%) |
| | | |
| Cumulative DOC yield | 1.40 g C m$^{-2}$ | 1.91 g C m$^{-2}$ |
| Spring DOC yield | 0.91 g C m$^{-2}$ (65%) | 1.11 g C m$^{-2}$ (58%) |
| Summer DOC yield | 0.49 g C m$^{-2}$ (35%) | 0.79 g C m$^{-2}$ (42%) |
| | | |
| Cumulative TDN yield | 40 mg N m$^{-2}$ | 51 mg N m$^{-2}$ |
| Spring TDN yield | 28 mg N m$^{-2}$ (69%) | 30 mg N m$^{-2}$ (58%) |
| Summer TDN yield | 13 mg N m$^{-2}$ (31%) | 22 mg N m$^{-2}$ (42%) |
| | | |
| Cumulative TP yield | 1.17 mg P m$^{-2}$ | 1.98 mg P m$^{-2}$ |
| Spring TP yield | 1.03 mg P m$^{-2}$ (89%) | 1.57 mg P m$^{-2}$ (79%) |
| Summer TP yield | 0.13 mg P m$^{-2}$ (11%) | 0.79 mg P m$^{-2}$ (21%) |
| | | |
| Cumulative TDP yield | 0.45 mg P m$^{-2}$ | 0.98 mg P m$^{-2}$ |
| Spring TDP yield | 0.39 mg P m$^{-2}$ (86%) | 0.85 mg P m$^{-2}$ (87%) |
| Summer TDP yield | 0.06 mg P m$^{-2}$ (14%) | 0.13 mg P m$^{-2}$ (13%) |

[1] April 29th to May 26th
[2] May 27th to September 9th

## 3.2 Water chemistry at catchment outlets

Stream water chemistry monitoring was initiated during the early stages of spring freshet, prior to complete ice-break-up as indicated by water temperatures near 0˚C during the first sampling occasions (Fig 4a). Stream water temperatures were

5   consistently 1.5˚C colder at the SC than NW outlet (Fig. 4a). Electrical conductivity increased from spring through summer, with minor decreases during summer storm events (Fig. 4b). The SC outlet had consistently lower Ec than the NW outlet during spring and until early July. Both SC and NW outlets exhibited a negative relationship between Ec and runoff (linear regression after log-log transformation: SC $R^2$ = 0.90, NW $R^2$ = 0.79), although Ec was lower than expected from this relationship during the rising limb of spring freshet (Fig 5b).

Both DOC and TDN concentrations reached seasonal minima during or just prior to peak spring runoff and then generally increased throughout the remainder of the study period (Fig. 4c,h). Concentrations of both DOC and TDN were negatively correlated with runoff at both outlets (linear regression after log-log transformation: SC $R^2$ = 0.71, NW $R^2$ = 0.89), but concentrations were lower during the rising limb of spring freshet than expected from these general relationships (Fig. 5b-c).

15   A strong correlation was found between DOC and TDN concentrations that was common for the two outlets (linear regression, $R^2$ = 0.91), and DOC/TDN mass ratios thus had a relatively low variability between 30 and 40 at both outlets. Concentrations of DOC and TDN were generally lower at the SC than NW outlet.

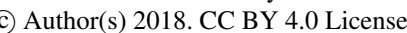



**Figure 4: Seasonal variation in catchment runoff and stream water chemistry measured at the outlets of Scotty Creek catchment and the recently burned Notawohka Creek catchment, NT, Canada. Showing: a) temperature (°C), b) electrical conductivity (µS cm$^{-1}$), c) DOC concentration (mg C L$^{-1}$), d) A$_{254nm}$ (cm$^{-1}$), SUVA (L mg C$^{-1}$ m$^{-1}$), f) S$_{275-295}$, g) total phosphorous (mg P L$^{-1}$), and total dissolved nitrogen (mg N L$^{-1}$). The shaded grey area indicates the spring period (Apr 29$^{th}$ – May 26$^{th}$).**



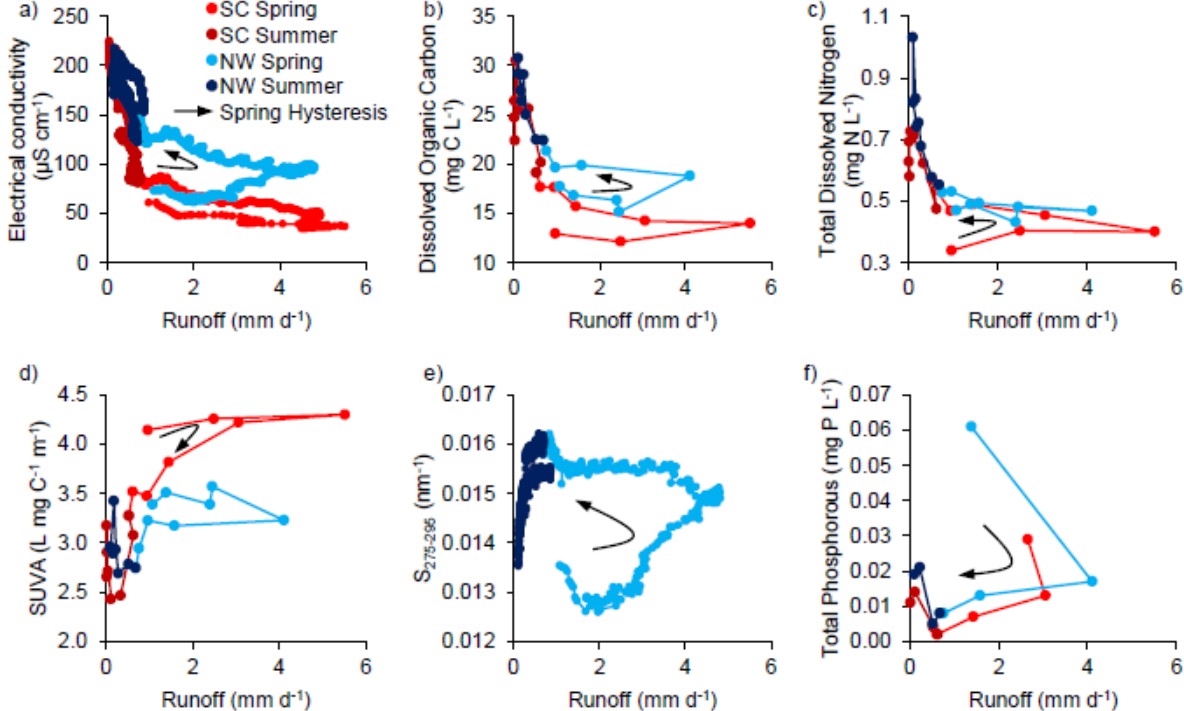

**Figure 5: Relationships between runoff and a) electrical conductivity, b) DOC concentration, c) total nitrogen, d) SUVA, e) $S_{275-295}$, f) total phosphorous at Scotty Creek (SC) and Notawohka Creek (NW) outlets in 2016. Spring (April 29th- May 26th) and summer periods (May 27th – Sep 9th) are indicated, and direction of hysteresis during spring is shown by arrows.**

While $A_{254}$ followed a seasonal pattern similar to DOC concentrations, with lowest absorbance during spring followed by generally rising absorbance throughout summer, we found that the ratio between $A_{254}$ and DOC, i.e. SUVA, varied significantly between 2.4 and 4.3 L mg $C^{-1}$ $m^{-1}$. Both SC and NW outlets had the highest SUVA, i.e. the highest DOC aromaticity, during the rising limb of spring freshet, followed by the lowest SUVA during early summer and then higher

10    again during late summer base flow conditions (Fig. 4e, 5d). The SC outlet had higher SUVA than the NW outlet during spring, while NW had higher SUVA than SC during most of the summer. We found an inverse relationship between SUVA and $S_{275-295}$ at the NW outlet ($R^2 = 0.64$, $p <0.01$), suggesting that DOC aromaticity and average molecular weight were positively correlated. The $S_{275-295}$ record from the NW outlet further emphasized the distinct, high molecular weight and high aromaticity, DOC characteristics during the rising limb of spring freshet (Fig. 5e).

Concentrations of TP and TDP were highly correlated ($R^2 > 0.95$, $p < 0.01$) and were highest during the rising limb of spring freshet, then decreased throughout spring, and rose slightly during low-runoff conditions in late summer (Fig. 4g). As such, TP and TDP were not correlated with runoff (Fig. 5f), or with DOC and TDN concentrations. However, TP and TDP concentrations were inversely correlated with $S_{275-295}$ at the NW outlet ($R^2 = 0.98$ and 0.97, respectively, both $p < 0.01$),





suggesting an association between high-molecular weight DOC and high TP/TDP during the rising limb of spring freshet, and during late summer base flow conditions.

### 3.3 Seasonal catchment DOC and nutrient yields

Cumulative DOC yield from the SC and NW catchments during the study period was 1.40 and 1.91 g C m$^{-2}$, respectively (Table 1). The lower DOC yield from the SC catchment was due to lower flow-weighted DOC concentration during spring (14.7 vs 18.2 mg C L$^{-1}$) when runoff was similar, and due to less runoff during summer (21 vs 31 mm) when concentrations were similar. The spring period dominated the DOC yield during the study period, with 58 and 65% of total DOC yield during spring from SC and NW, respectively. The rising limb of spring freshet, which had distinct DOC characteristics with regards to SUVA and $S_{275-295}$, was responsible for 27 and 26% of the total DOC yield from SC and NW, respectively. Yield of TDN, at 40 and 51 mg N m$^{-2}$ respectively from SC and NW, was proportional to DOC yield throughout the study period with DOC/TDN mass ratios of 35 and 37 from SC and NW, respectively. Yield of TP was lower from the SC than NW catchment, at 1.17 and 1.98 mg P m$^{-2}$, respectively, mainly due to a flow-weighted TP concentration that was lower at the SC than NW outlet, at 0.014 and 0.022 mg P L$^{-1}$. The DOC/TP yield mass ratio was overall greater at the SC than NW outlets, at 1200 and 960, respectively. The TP yields were even more dominated by the spring period than DOC and TDN yield, with 79 and 89% of the total TP yield from SC and NW catchments occurring during the spring period, and > 50% of the total TP yield occurring during the rising limb of spring freshet at both catchment outlets.

We estimated historical DOC yield between 1995 and 2015 for the SC catchment using the historical runoff record and the relationship between runoff and DOC concentrations in 2016. We found that large inter-annual variability in cumulative runoff for the period from onset of the spring freshet until September 9$^{th}$ (125 ±57 mm ±1SD) (Fig. 3) led to highly variable DOC yield (2.2 ±0.9 g C m$^{-2}$). In comparison to the long term average, 2016 was estimated to have 50% greater DOC yield during the spring period but only a third of the average DOC yield during summer, with overall lower than average DOC yield.

### 3.4 Streamflow contribution from different water sources

Both catchments exhibited similar seasonal shifts in Ec and $A_{254}$, shifts which stayed within the mixing-space of the three identified potential end-members – precipitation-like water, mineral soil groundwater, and organic soil porewater (Fig. 6a and b). The end-member mixing model indicated that streamflow was dominated by precipitation-like water during the spring period, with a shift towards mineral soil groundwater during summer. The continued contribution of precipitation-like water outside the freshet and storm events may indicate release of stored precipitation-like water in lakes and wetlands. The organic soil porewater contribution showed the least variability of the three end-members over the season, contributing 15-20% of streamflow during spring and 20-30% of streamflow during summer.





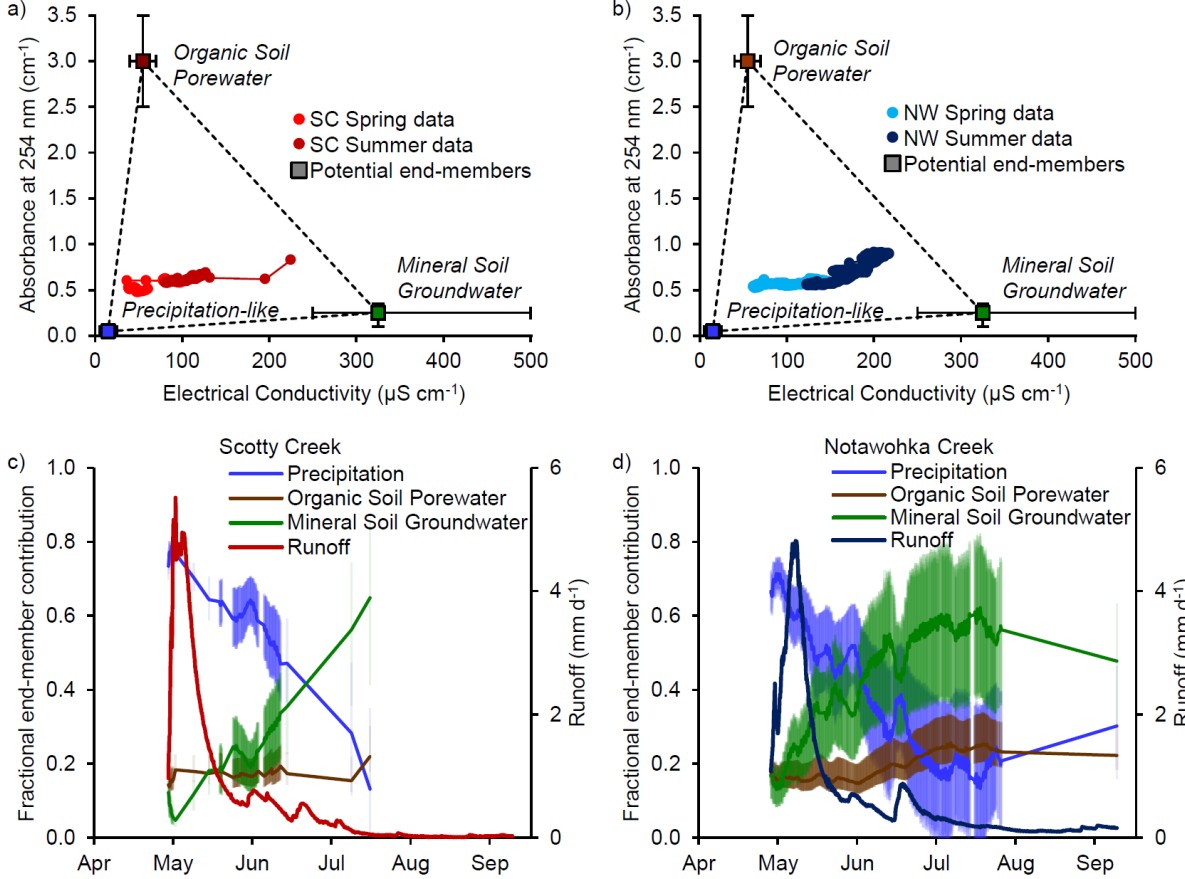

**Figure 6: End-member mixing space defined by decadal absorbance at 254 nm and electrical conductivity with outlet stream data from a) Scotty Creek and b) Notawohka Creek, and the fractional contribution to streamflow from potential end-members at c) Scotty Creek and d) Notawohka Creek catchment outlets. Error bars indicate the 95% CI for fractional end-member contribution to streamflow.**

### 3.5 Peatland porewater chemistry

Wildfire was found to increase porewater TDP concentrations on the peat plateau, and to increase the aromaticity of the DOC (higher SUVA), but was not found to influence concentrations of DOC or TDN (Fig. 7). Thermokarst bogs, which are unaffected by wildfire, had porewater characteristics similar to unburned peat plateau, with the exception of lower TDN concentrations. From May until September across the three sampling occasions, there was a general trend of increasing $A_{254}$ (from 1.8 to 4.0 m$^{-1}$, on average across all sites), DOC (from 52 to 100 mg C L$^{-1}$), TDN (from 1.2 to 1.8 mg N L$^{-1}$), and SUVA (from 3.5 to 4.0 L mg C$^{-1}$ m$^{-1}$). Concentrations of TDP were stable during the study period at the unburned peat plateau and thermokarst bog but decreased at the burned peat plateau (from 0.61 to 0.24 mg P L$^{-1}$).





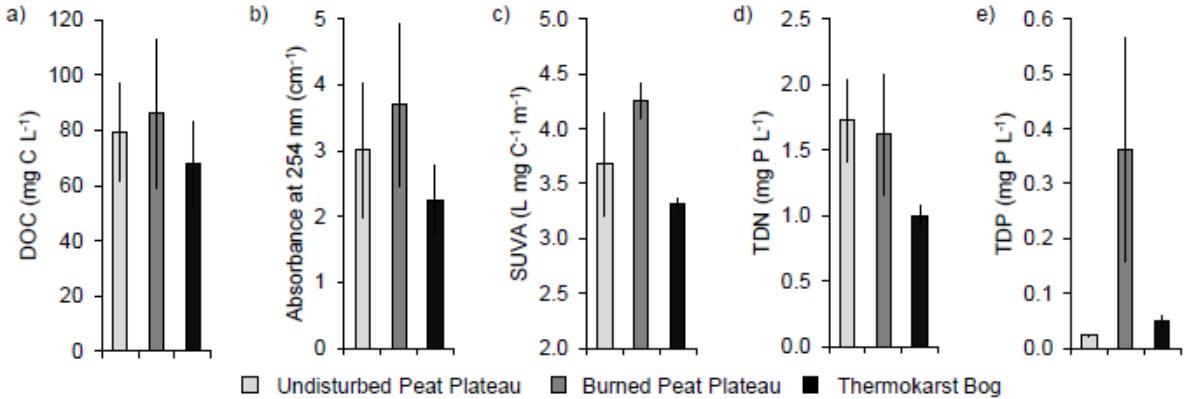

**Figure 7: Porewater characteristics at undisturbed peat plateaus, recently burned peat plateaus, and thermokarst bogs; a) dissolved organic carbon (DOC), b) decadal absorbance at 254 nm, c) Specific UV absorbance at 254 nm (SUVA), d) total dissolved nitrogen (TDN), and e) total dissolved phosphorous (TDP). Error bars indicate 1 SD based on three samples collected in early June, late July, and early September. The partially burned peatland site with both undisturbed and burned peat plateaus is located just outside the Notawohka catchment.**

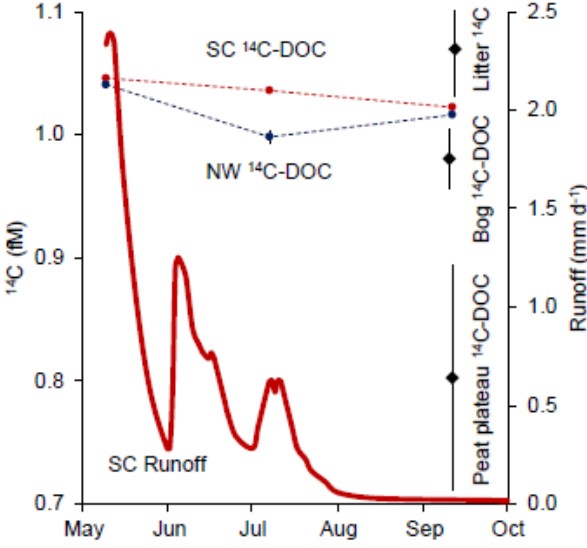

**Figure 8: Radiocarbon characteristics of DOC at the Scotty Creek (SC) and Notawohka Creek (NW) outflows, of DOC in peat plateau (burned and unburned) and thermokarst bog pore water, and of forest floor litter. The Scotty Creek hydrograph for 2017 is shown. Error bars indicate 1 SD among replicate samples. Error bars for stream DOC are all <0.005 and thus hard to see.**

### 3.6 Radiocarbon dating

Peat plateau porewater had significantly aged DOC, but showed no effect of wildfire, as both burned and unburned peat plateau DOC had an average fM of 0.80 (Fig. 8). In contrast, thermokarst bog porewater had only a minor aged DOC component at fM 0.98. Forest floor litter from upland mixed-wood forests, thermokarst bogs, and peat plateaus were strongly



influenced by $^{14}$C bomb enriched DOC with an average fM of 1.07. Stream water DOC at the SC and NW outlets varied

between 1.05 and 0.99, with the SC having higher values at each of the three sampling occasions – particularly in July (Fig.

8).

## 4 Discussion

5   This study found overall similar seasonal patterns in DOC and nutrient yield from a recently burned (3 years prior) and an

unburned catchment on the peatland-rich Taiga Plains in the discontinuous permafrost zone of western Canada. A few

differences between the two catchments were consistent with impacts of fire on peatland porewater characteristics. Below we

emphasize key results from this study that help us understand the controls on stream chemistry during transitions from

periods of high to low runoff, the importance of spring freshet for the annual solute yield, the impacts of wildfire on solute

10   yield, and the potential for climate change to affect the magnitude of catchment DOC and nutrient yields from peatland-rich

catchments in a region where similar studies are lacking.

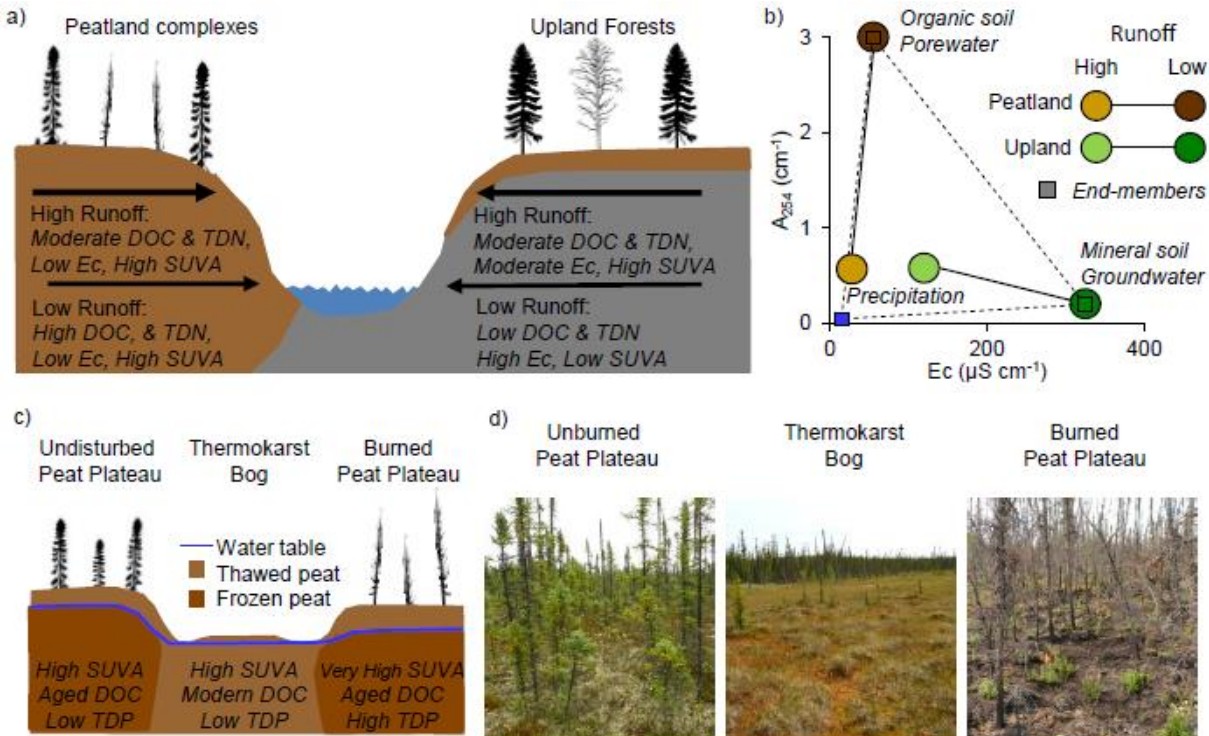

**Figure 9. Conceptual controls on seasonal variation in stream chemistry for heterogeneous catchments with mixed peatland and upland forest land cover on the Taiga Plains of western Canada. a) General water characteristics of runoff from peatlands and upland forests during low and high runoff generation, respectively. b) Water chemistry of peatland and upland forest runoff during high and low runoff periods, as conceptualized as a mixing of three end-members; organic soil porewater, precipitation-like water, and mineral soil groundwater. Dotted line indicates the mixing space of end-members. c) Differences in DOC and nutrient characteristics among thermokarst bogs, undisturbed peat plateaus, and burned peat plateaus. d) Representative photos of the three peatland types.**





### 4.3 Shifts in stream water chemistry as related to water sources

Seasonal shifts in water chemistry and DOC characteristics at catchment outlets are common, but patterns vary strongly among catchments (Fellman et al., 2009; Larouche et al., 2015; Vonk et al., 2015; Broder et al., 2017). We found that the variability in water chemistry and DOC characteristics in this study was largely consistent with a model for runoff generation

and solute yield from peatlands and upland forests based on data from boreal northern Europe, despite distinct differences in surficial geology and permafrost conditions (Laudon et al., 2011, Winterdahl et al., 2011). In this model, rising peatland runoff due to snowmelt and rain cause dilution of both DOC and TDN concentrations, while increasing runoff from upland forests shifts dominant riparian flow paths from deeper mineral soils to shallow organic soils and thus leads to decreasing concentrations of most dissolved ions but increasing DOC and TDN concentrations (Fig 9a). These patterns of shifting water

chemistry in runoff from peatland and upland forests can be conceptualized to result from mixing of the three end-members used in our analysis (Fig. 9b). In runoff from peatlands, organic soil porewater would be expected to dominate under low-flow periods, and to have an increasing contribution from precipitation-like water under high-flow periods. Runoff from upland forests would be expected to be dominated by mineral soil groundwater during low-flow periods, with the contribution from both organic soil pore-water and precipitation-like water expected to increase under high-flow periods.

Overall, water chemistry at catchment outlets is thus expected to be a function of runoff conditions and the relative abundance of wetland and upland forests within the catchment.

We accordingly found that DOC and TDN concentrations increased as runoff decreased after the spring period, which is consistent with peatlands dominating DOC and TDN yield from boreal catchments with >60% peatlands cover and the

dilution from precipitation-like water during high flow periods (Laudon et al., 2011). The strong link between catchment DOC and TDN yield is common for boreal catchments (Kortelainen et al., 2006). The concurrent increase in Ec after the spring period was however likely unrelated to shifting characteristics of peatland runoff, but rather linked to the parallel increase in the relative contribution from mineral soil groundwater sources to streamflow. The continued contribution from precipitation-like water well into the summer may be related to slow release of snowmelt from storage in peatlands and

lakes, as indicated by previous isotopic separation techniques for Scotty Creek (Hayashi et al., 2004).

The mixing between precipitation-like water, organic soil porewater, and mineral soil groundwater can also explain the seasonal variation in DOC aromaticity as indicated by SUVA. The aromaticity of DOC with high concentrations in peatland porewater in this study was similar to other studies with SUVA in the range 3.0 to 4.0 L mg $C^{-1}$ $m^{-1}$ (Tfaily et al., 2013;

O'Donnell et al., 2016). This contrasts to the SUVA of low concentration, microbially derived DOC found in mineral soil groundwater, which is often <1.0 L mg $C^{-1}$ $m^{-1}$ (Olefeldt et al., 2013a; Shen et al., 2015). Stream water DOC thus had SUVA similar to peatland porewater during the spring period, albeit diluted by snowmelt, while stream DOC during summer had SUVA between 2.5 and 3.0 L mg $C^{-1}$ $m^{-1}$ which indicates a minor contribution also from mineral soil groundwater DOC.





Despite low aromaticity, microbially derived DOC in mineral soil groundwater has been found to have low microbial lability (Olefeldt et al., 2013a) and thus the shift in stream water DOC characteristics from spring to summer likely indicate reduced microbial lability (Wickland et al., 2012). Although the mixing model helps explain much of the seasonal variability in Ec, TDN, DOC and DOC characteristics, patterns in TDP concentrations did not follow shifts in the relative contribution of the

three identified end-members – particularly during the rising limb of spring freshet.

## 4.2 Importance of spring for catchment solute yield

This study emphasizes the need to characterize solute yield accurately during spring when studying northern catchments, and during early spring in particular despite practical challenges (Holmes et al., 2012). Catchment runoff from both catchments during the study period in 2016 was evenly distributed between the rising limb of spring freshet (lasting ~10 days), falling

limb of spring freshet (~18 days), and the summer period (~105 days). Similarly, >25% of catchment yield of DOC and TDN occurred during the rising limb of freshet, and another 25% during the falling limb, while >50% of the TP and TDP yields occurred during the rising limb of spring freshet, with another 35% during the falling limb. Dominance of spring freshet for the annual DOC and TDN yield is common for boreal and subarctic catchments (Finlay et al., 2006; Dyson et al., 2011; Olefeldt and Roulet, 2014), but the importance of TP yield during spring in this study was greater than observed elsewhere

(Eimers et al., 2009).

The distinct stream chemistry during the rising limb of spring freshet was likely associated with contribution to streamflow from flow-paths that are only hydrologically connected during this period, e.g. surficial flow-paths forced by frozen ground (Ågren 2008). The rising limb of snowmelt was associated with the highest DOC aromaticity (highest SUVA) of the study

period and the greatest DOC molecular weight (lowest $S_{275-295}$). Previous studies of Scandinavian catchments with significant wetland coverage found conversely that the spring period had DOC with lower aromaticity than during the rest of the year (Ågren et al 2008; Olefeldt and Roulet, 2014), while wetland catchments in Alaska had similar results as in this study (O'Donnell et al. 2010). The reason for these differences is not clear, but could possibly be due to ecosystem-specific DOC characteristics of near-surface soil porewater. Several studies have found that DOC yielded during spring is of higher

microbial lability than during summer (Mann et al., 2012; Wickland et al., 2012), which further emphasizes the importance of spring DOC yield for downstream biogeochemistry.

The rising limb of freshet was also associated with the highest TP and TDP concentrations during the study period. Yield of TP and TDP from other boreal headwater catchments have been considered to be primarily as organic P, as indicated by

strong associations between DOC and TP/TDP concentrations (Dillon and Molot 1997; Eimers et al. 2009).  In this study we had no correlation between DOC and TP/TDP concentrations, and the highest TP/TDP concentrations in spring occurred when the streams were noticeably turbid. This finding suggested that spring export of inorganic P, e.g. in the form of



phosphate bound to calcium or iron in colloidal particles (Reddy et al. 1999; Wang et al 2005), was a major contributor to the annual catchment P yield.

## 4.3 Impacts of wildfire on catchment solute yield

The greater TP and TDP yields from the burned Notawohka catchment than from the undisturbed Scotty Creek catchments

was likely due to effects of fire. Fire leads to mineralization of organic P, but not to gaseous combustion losses as for C and N (Neff et al., 2005), and this explains the observed higher TDP concentrations in porewater on the burned peat plateau compared to the unburned peat plateau. Increased catchment yield of TP or TDP following fire has previously been indicated for several other Canadian regions, including the non-permafrost boreal plains in western Canada (McEachern et al. 2000; Burke et al., 2005), the boreal shield of eastern Canada (Lamontagne et al., 2000), and in the foothills of the Rocky

Mountains (Silins et al., 2014). This study suggests that the ratio between annual cumulative DOC and TP yield may be a sensitive indicator of effects of wildfire when comparing across boreal catchments under different climates. In this study we found yield ratios of DOC to TP yield at 1200 and 950 for the undisturbed and burned catchments, respectively. This is consistent with other boreal catchments where the range has been 1200 to 2000 for undisturbed and <1000 for burned catchments (Dillon and Molot, 1997; Lamontagne et al., 2000; Kortelainen et al., 2006). Increased TP yield has been linked

to increased stream algal production, suggesting much of the additional P loading is reactive, which in turn has cascading effects on high trophic levels, e.g. on invertebrate and fish populations (Silins et al., 2014). Qualitatively, we observed noticeably greater epiphytic algal growth in the stream channel of Notawohka Creek than Scotty Creek, suggesting that increased catchment P yield following fire in the study region may have important effects on aquatic productivity and food web structure.

The effect of wildfire on catchment yield of DOC has been inconsistent among studies of boreal catchments (Lamontagne et al., 2000; McEachern et al 2000; Petrone et al., 2007; Olefeldt et al 2013a; Parham et al., 2013). This suggests that climate or catchment characteristics are likely to modulate any effects of wildfire. In this study, we observed greater DOC yield from the burned Notawohka Creek catchment during summer, a period when DOC yield further was associated with higher DOC

aromaticity and a greater contribution from aged DOC than from the undisturbed Scotty Creek catchment. While this could arise from many factors which this study cannot differentiate, this difference is consistent with increased runoff generation from burned peat plateaus during summer. Peat plateaus are slightly raised above the surrounding peatland, and thus shed water throughout summer as the seasonally thawed layer deepens (Quinton et al., 2009). The seasonally thawed layer of peat plateaus in the study region increases from a maximum of ~70 cm to ~120 cm during the first few years following fire due to

altered surface energy balance (Gibson et al., In review), which thus creates an increased potential for summer runoff generation. As the burned peat plateau had porewater with higher DOC aromaticity and higher TDP concentrations than the undisturbed peat plateau (Fig. 9c-d), increased summer runoff generation after fire would thus be expected to lead to the observed differences at the catchment outlets.





The increased contribution of aged DOC to catchment DOC yield during summer is also consistent with increased contribution of runoff from burned peat plateaus to catchment DOC export. Both the burned and unburned peat plateaus were found to have significantly aged porewater DOC (fM: 0.65-0.85: 3,700-1,250 cal BP), while the non-permafrost bogs had much younger porewater DOC (fM: 0.95-1.00, 500-0 cal BP) (Fig. 9c). Several other boreal non-permafrost peatlands have similarly been found to have predominately modern porewater DOC, even at several meters depth where the peat itself is thousands of years old (Wilson et al. 2016; Campeau et al. 2017). An important difference between peat plateaus and boreal non-permafrost peatlands may be the greater depth of the aerobic surface layer in peat plateau systems (Fig. 9c) that could enhance DOC production from aged peat. In contrast, peat under anaerobic conditions in wetter non-permafrost bogs appears largely inert, and DOC at significant depth has been found to be modern (fM ~1.0) - likely leached from plants near the surface (Wilson et al. 2016; Campeau et al. 2017). Despite the aged DOC in peat plateau porewater, DOC at the catchment outlets was still found to be predominately modern (fM: 0.99-1.05). The sampling in July at the Notawohka Creek outlet was in fact the only occasion when the stream DOC sample was not dominated by modern C (fM <1.00). This suggests that while catchment DOC yields were mainly comprised of young or bomb-peak influenced DOC, e.g. the thermokarst bog DOC porewater (fM: 0.95-1.00) or from DOC leachates from forest litter (fM: 1.05-1.10), there may have been an additional contribution from the burned peat plateaus in the NW catchment.

### 4.4 Climatic controls on DOC and nutrient yield on the Taiga Plains

The dry climate of the study region restricted the cumulative catchment DOC yield to < 2 g C m$^{-2}$ for the study period, which is substantially lower than the range 4 to 15 g C m$^{-2}$ yr$^{-1}$ found for boreal catchments in other regions with similar peatland coverage (Lamontagne et al. 2000; Olefeldt et al. 2013b). Runoff during the 2016 study period from the Scotty Creek catchment was 85 mm, below the long term (1995-2015) average of 125 mm for the same period. However, the long term record also shows that the region has a very large inter-annual variability in runoff generation, with a range in annual runoff between 30 and 330 mm. If we assume that the relationship between runoff and DOC concentration (Fig. 5b) has remained steady for this period, we can estimate that the annual DOC yield has varied between 0.6 and 5.0 g C m$^{-2}$ yr$^{-1}$. The large inter-annual variability in runoff and DOC yield is linked to the balance between precipitation and evapotranspiration in the region, which means that small changes to either will lead to relatively large changes in runoff. Hence, the catchment yield of DOC and nutrients in the discontinuous permafrost zone of the Taiga Plains is likely relatively much more sensitive than other boreal regions to climate change.

Climate change may also influence catchment DOC and nutrient yield in the study region through permafrost thaw. Permafrost thaw in peatlands in this study region is associated with a transition of peat plateaus into thermokarst bogs and channel fens (Chasmer and Hopkinson, 2017; Gibson et al., In review), a transition which alters both the porewater DOC characteristics (Gordon et al., 2016), and the landscape hydrological connectivity (Connon et al., 2015). Variability in mean





annual runoff among catchments in this region has been linked to the relative abundance of channel fens (Quinton et al. 2011), and increasing runoff from Scotty Creek during the period 1997 to 2011 has been linked to ongoing loss of peat plateaus (Chasmer and Hopkinson, 2017). The transition from the Taiga Plains, where peatlands have discontinuous permafrost, to the Boreal Plains further south where permafrost is absent has been shown to correspond to significantly

higher stream DOC concentrations when comparing catchments with similar wetland extents (Olefeldt et al., 2014). Continued permafrost thaw is thus expected to increase catchment DOC yield and alter its chemical characteristics (Wauthy et al., 2018). On the Taiga Plains this effect of climate change on catchment solute yield characteristics is further likely to be accelerated by wildfire, as wildfire is a dominant disturbance and has been found to significantly accelerate the  rate of peat plateau loss due to permafrost thaw (Gibson et al., In review).

**5 Conclusions**

In this study we have shown that monitoring during the traditionally understudied spring freshet, particularly the rising limb of freshet, is required to accurately characterize catchment DOC and nutrient yield. In the study year, more than half of cumulative TP yield occurred during the first week of freshet, and more than half of the DOC and TN yield occurred during the four week spring period. The burned catchment was found to have a significantly higher TDP yield than the undisturbed

catchment, and this difference appeared linked to increased availability of TDP in porewater of burned ecosystems. Effects of wildfire on catchment DOC and TN yield were less clear, although increased DOC yield during summer, with greater aromaticity and greater contribution from aged C, was consistent with increased runoff generation during summer from burned peat plateaus where the seasonally thawed soil layer is rapidly deepening during the first few years after fire. Further studies are however required to link wildfire definitively to these effects on DOC and TN yield in the study region. Our

results suggest that the effects of wildfire on catchment DOC and TN yield are likely to be less important than expected changes anticipated from climate change, due to its effects on permafrost thaw and runoff generation.

**Data Availability**

The data that support the findings of this study are available from the corresponding author upon reasonable request.

**Author contribution**

KB, DO, and SET conceived and designed the study. KB, DO, and ND collected data. AJT and SET facilitated the radiocarbon analysis. KB, and DO carried out the data analysis. All authors aided in data interpretation and the writing of the manuscript.



**Competing interests**

The authors declare no competing financial interests.

**Acknowledgments**

This study was funded by support from the National Science and Engineering Research Council Discovery grant (RGPIN-
5    2016-04688), the Campus Alberta Innovates Program, the University of Alberta Northern Research Awards, a UK-Arctic
Canada Arctic Partnership bursary from the Department for Business, Energy and Industrial Strategy supported by the
NERC Arctic Office, and Polar Knowledge Canada (POLAR) Science and Technology program. We thank William
Heffernan, Carolyn Gibson, Michael Barbeau, Jessi Steinke, Megan Schmidt, Cristian Estop-Aragones, and McKenzie Kuhn
for their help with field work.



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
