# Peer review of "Seasonal shifts in export of DOC and nutrients from burned and unburned peatland-rich catchments, Northwest Territories, Canada"

_Hydrology and Earth System Sciences, 2018_

## Referee Comment (RC1) · Anonymous Referee #1 · 13 Jun 2018

I recommend this paper for publication with minor revisions. This paper was enjoyable and provided a good comparison across DOC, N and P yields from a disturbed and undisturbed catchments with ∼60% peatland cover using several different lines of evidence and analyses. The authors used fluorescence indices, 14C dating of DOC, DOC, P and N to assess quantity (yields) and composition (DOC quality) of aqueous export from each catchment. The authors highlight the importance of catchment dynamics using end-member analysis, hysteresis, radiocarbon dating and by quantifying runoff and solute yields. Monitoring during spring freshet were shown as critically important to accurately characterize DOC and nutrient yields. The final conclusions stated in the manuscript suggest that climate change will alter DOC composition, and

[Figure]

DOC and TN yield more than wildfire. Although the study is interesting and illuminating in many respects, sentences are long and sometimes difficult to follow. Making some sentences (a few indicated below) more concise and targeted would clarify important messages for the reader. There are no details about the fire in the disturbed Notawohka peatland (areal extent, severity/depth of burn) and this information would help contextualize the effects of wildfire on catchment yields. The omission of details about the specifics of wildfire occurrence in the Notawohka catchment is striking and should be included in the manuscript if this catchment is being directly compared to an 'undisturbed' catchment in each analysis. Specific comments: P3, L7-8: "Immediacy of links between terrestrial and aquatic biogeochemistry..." could be rewritten for clarification P3, L10-11: Same as above. Very important message but lacks clarity. P3, L16: "generally cause increased..." – May not be the best word. P3, L17-19: Awkward phrasing. Unclear. P5, L2: "mesic in wetness"? Page 5, L5: What does "carry fire well" mean? Page 7, L6: Reference for stannous chloride method Page 9, L5: What percentage of the peatland complex? Page 9, L21: "...stored dark and cool" – Odd phrase. Page 12: Figures a), b), d), f) – Difficult to decipher symbols. Page 15: b) – is it possible to get some transparency because superposition of data points blocks data pattern c) & d) the shading is confusing. Technical corrections (typos): Page 2, L24: change order of references according to date to coincide with rest of paper Page 3, L7: mean to means (typo), possibly change "...contributing sources.." to "...sources contributing to..." Page 9, L8: Remove "," after "stream water samples..." Page 9, L16: Add colon after "occasions in 2017:..." Page, L18: Add period after "for 4 h). " Page 9, L21: "stored dark and cool" sounds awkward. Page 3, L22: remove "," in the sentence "...the permafrost, i.e. the active layer, (Gibson.." add comma (Gibson et al., In review)," Page 3,L23: Add "a" after "...this region may exhibit a characteristic response..." Page 10, L15: Reference for Scotty Creek catchment weather could be improved. Page 14, L25: Remove 'shifts" after A254 ?
* * *
253, 2018.

---

## Referee Comment (RC2) · Anonymous Referee #2 · 20 Jun 2018

General comment: In this study, researchers from Canada qualitatively characterized DOC, and compared yields of DOC and nutrients from two boreal peatland catchments from early spring to autumn. The two peatland catchments differed in terms of flow and disturbance regimes (wildfire). Major findings of this study are that 1) the two catchments showed strong similar DOC and nutrient exports. 2) more than half of the exported DOC and phosphorus occurred during spring, with the rising limb of the freshet linked to higher phosphorus concentrations and DOC of higher molecular weight. 3) the burned catchment had significantly increased total phosphorus and DOC yield. The main conclusion of the study was that predicted changes in runoff may be more important for the DOC character and export from boreal peatlands than wild fires. This is an

interesting and important study, and the manuscript is in very good shape. I strongly recommend this manuscript to be published with minor revisions.

Questions: 1-Page 4, lines 10, 11 and 12- "The SC outlet at the Liard Highway (61°24 W, 121°26 N) has a 134 km2 10 catchment that has not been affected by any major fires in the last 60 years. The NW outlet at the Mackenzie Highway (61°08 W, 120°17 N) has a 321 km2 catchment that was >90% burned in 2013 (Northwest Territories Fire Scar Map, 2013)." Is it possible that difference in catchment size affected the export of DOC and nutrients from peatlands? Typically, large forested catchments drain deeper soil layers, resulting in lower exports of DOC and TDN. Therefore, could catchment size influence the DOC and nutrient yields from peatland catchments?

2-This comment is also related to catchment size: is it possible that differences in catchment size influenced the DOC yields (due to microbial respiration during different water transit times)?

Technical corrections/suggestions: 1-Optional idea: Page 8, line 5- "(R2 = 0.71, p < 0.005)." and so forth. -You may add sample size.

2-Page 8, line 5- Replace estimated by estimates.

3-Page 18, line 1- Replace 4.3 by 4.1.

---

## Author Comment (AC1) · 12 Jul 2018

We thank the two referees for their comments, which have allowed us to further improve a few key points and overall clarity of the manuscript. Below we outline our answers to both referees, with suggested changes indicated.

**Anonymous Referee #1**

**I recommend this paper for publication with minor revisions. This paper was enjoyable and provided a good comparison across DOC, N and P yields from a disturbed and undisturbed catchments with ~60% peatland cover using several different lines of evidence and analyses. The authors used fluorescence indices, 14C dating of DOC, DOC, P and N to assess quantity (yields) and composition (DOC quality) of aqueous export from each catchment. The authors highlight the importance of catchment dynamics using end-member analysis, hysteresis, radiocarbon dating and by quantifying runoff and solute yields. Monitoring during spring freshet were shown as critically important to accurately characterize DOC and nutrient yields. The final conclusions stated in the manuscript suggest that climate change will alter DOC composition, and DOC and TN yield more than wildfire.**

**Although the study is interesting and illuminating in many respects, sentences are long and sometimes difficult to follow. Making some sentences (a few indicated below) more concise and targeted would clarify important messages for the reader**.

Our response: We have addressed all comments below, and will also go through the overall text to improve sentence clarity and brevity.

**There are no details about the fire in the disturbed Notawohka peatland (areal extent, severity/depth of burn) and this information would help contextualize the effects of wildfire on catchment yields. The omission of details about the specifics of wildfire occurrence in the Notawohka catchment is striking and should be included in the manuscript if this catchment is being directly compared to an 'undisturbed' catchment in each analysis.**

Our response: The fire burned between June 29 and August 20[th], 2013, reaching an area of 967 km$^2$, affecting ~90% of the Notawohka catchment. Since the fire burned over an extended period of time, there will have been highly variable weather conditions during this time affecting fire intensity. No fire severity analysis has been made of this fire in particular, but recent studies of depth of burn in the region has shown that there is substantial variability both between different ecosystems and within individual ecosystems types – even within a single fire scar (Walker et al., Global Change Biology, In press – available as Early View). Our field observations were limited to burned areas accessible by foot from the road, where we observed indicators of relatively low fire severity in the peat plateau that we visited. However, we can not determine whether this is representative of the fire severity throughout the catchment. We have added these sentences to the catchment description in the methods section:

*The NW outlet at the Mackenzie Highway (61°08 W, 120°17 N) has a 321 km$^2$ catchment that was >90% burned in 2013 (Northwest Territories Fire Scar Map, 2013). The fire burned a total area of 967 km$^2$*

*between late June and mid-August, thus likely having variable fire severity within the fire scar (Xanthe et al., 2018).*

Specific comments:

**P3, L7-8: "Immediacy of links between terrestrial and aquatic biogeochemistry. . ." could be rewritten for clarification**

Our response: Sentence has been changed to: *However, DOC radiocarbon age further indicates whether links between terrestrial and aquatic biogeochemistry predominately act on short or long time scales.*

**P3, L10-11: Same as above. Very important message but lacks clarity.**

Our response: Sentence has been changed to: *Shifts in stream DOC composition may thus be as important for downstream ecosystems as a shift in catchment DOC export magnitude, and both aspects of catchment DOC export are potentially influenced both directly by climate change and indirectly e.g. though impacts of wildfire.*

**P3, L16: "generally cause increased. . ." – May not be the best word.**

Our response: Sentence has been changed to: *While fire is generally found to cause increased catchment export of total phosphorous in the boreal biome, observed impacts of fire on DOC and total dissolved nitrogen export include increases, decreases, and no change.*

**P3, L17-19: Awkward phrasing. Unclear.**

Our response: Sentence has been changed to, same as above: *While fire is generally found to cause increased catchment export of total phosphorous in the boreal biome, observed impacts of fire on DOC and total dissolved nitrogen export include increases, decreases, and no change*

**P5, L2: "mesic in wetness"?**

Our response: Sentence has been changed to: *Peat plateaus are relatively dry and dominated by black spruce (Picea mariana), Labrador tea (Rhododendron groenlandicum), and a variety of lichen species; thermokarst bogs have a water table 5 to 40 cm below the surface and support Sphagnum spp mosses and low shrubs, while channel fens have a persistent water table above the soil surface and vary from being dominated by sedges and other tall graminoids to being dominated by shrubs mostly from the genus Betula*

**Page 5, L5: What does "carry fire well" mean?**

Our response: Sentence has been changed: *Neither thermokarst bogs nor channel fens burn readily well as they are wetter and lack trees, hence these ecosystems were largely unaffected by the fire that burned the Notawohka Creek catchment.*

**Page 7, L6: Reference for stannous chloride method**

Our response: Sentence has been changed: *An additional 8 samples were analysed photometrically (690 nm) for total phosphorous (TP) and total dissolved phosphorous (TDP) concentrations through stannous chloride method (Standard Method 4500-P:D).*

**Page 9, L5: What percentage of the peatland complex?**

Our response: Should not be relevant to results. Perhaps two thirds of this peatland complex was affected by fire. This peatland complex is ~3 km$^2$ large and just one of many in the region. No change made.

**Page 9, L21: ". . .stored dark and cool" – Odd phrase.**

Our response: Explanation added: *Pre-combusted bottles (0.5 L) were filled at each pit and then stored dark and cool to avoid photochemical or microbial degradation of DOC.*

**Page 12: Figures a), b), d), f) – Difficult to decipher symbols.**

Our response: Legends have been altered to more clearly show symbols.

**Page 15: b) – is it possible to get some transparency because superposition of data points blocks data pattern c) & d) the shading is confusing.**

Our response: We tried changing transparency in a) and b), but it does not improve clarity of the figures. The take-home message is that water type shifts from precip-dominated to groundwater-dominated, and the current style shows that. No change made. Shading in c) and d) is actually error-bars that indicate the 95% CI for the end-member mixing model for each half hour data, when all necessary data is available. We made changes to the figure legend to improve clarity of what the error bars indicate: *Error bars in a) and b) indicate the 95% CI of the end-member characteristics (see Methods for justification), while error bars in c) and d) indicate the 95% CI for fractional end-member contribution to streamflow based on Eqs. 1 – 3 and the uncertainty of the end-member characterization.*

**Technical corrections (typos):**

**Page 2, L24: change order of references according to date to coincide with rest of paper**

Our response: Done.

**Page 3, L7: mean to means (typo), possibly change ". . .contributing sources.." to ". . .sources contributing to. . ."**

Our response: Sentence has been changed to: *Different catchment DOC sources may also vary in terms of radiocarbon ($^{14}$C) age (Raymond et al., 2007), and thus provides another means to differentiate between sources contributing to catchment DOC export.*

**Page 9, L8: Remove "," after "stream water samples. . ."**

Our response: Done.

**Page 9, L16: Add colon after "occasions in 2017:. . ."**

Our response: Done.

**Page, L18: Add period after "for 4 h). "**

Our response: Done.

**Page 9, L21: "stored dark and cool" sounds awkward.**

Our response: Sentence changed: *Pre-combusted bottles (0.5 L) were filled at each pit and then stored dark and cool to avoid photochemical or microbial degradation of DOC.*

**Page 3, L22: remove "," in the sentence ". . .the permafrost, i.e. the active layer, (Gibson.." add comma (Gibson et al., In review),"**

Our response: Done.

**Page 3,L23: Add "a" after ". . .this region may exhibit a characteristic response. . ."**

Our response: Done.

**Page 10, L15: Reference for Scotty Creek catchment weather could be improved.**

Our response: Reference changed to: (*Data available through the Government of Canada, Environment and Climate Change Canada: climate.weather.gc.ca*)

**Page 14, L25: Remove 'shifts" after A254 ?**

Our response: Done.

**Anonymous Referee #2**

**General comment: In this study, researchers from Canada qualitatively characterized DOC, and compared yields of DOC and nutrients from two boreal peatland catchments from early spring to autumn. The two peatland catchments differed in terms of flow and disturbance regimes (wildfire). Major findings of this study are that 1) the two catchments showed strong similar DOC and nutrient exports. 2) more than half of the exported DOC and phosphorus occurred during spring, with the rising limb of the freshet linked to higher phosphorus concentrations and DOC of higher molecular weight. 3) the burned catchment had significantly increased total phosphorus and DOC yield. The main conclusion of the study was that predicted changes in runoff may be more important for the DOC character and export from boreal peatlands than wild fires. This is an interesting and important study, and the manuscript is in very good shape. I strongly recommend this manuscript to be** Our response:

**published with minor revisions.**

Questions:

**1-Page 4, lines 10, 11 and 12- "The SC outlet at the Liard Highway (61◦24 W, 121◦26 N) has a 134 km2 10 catchment that has not been affected by any major fires in the last 60 years. The NW outlet at the Mackenzie Highway (61◦08 W, 120◦17 N) has a 321 km2 catchment that was >90% burned in 2013 (Northwest Territories Fire Scar Map, 2013)." Is it possible that difference in catchment size affected the export of DOC and nutrients from peatlands? Typically, large forested catchments drain deeper soil layers, resulting in lower exports of DOC and TDN. Therefore, could catchment size influence the DOC and nutrient yields from peatland catchments?**

Our response: Yes, catchment size may influence catchment DOC export, largely thought to be due to increased DOC losses within aquatic (streams/rivers/lakes) ecosystems due to microbial and photochemical losses. Usually this also includes a selective removal of aromatic DOC due to flocculation and photochemical transformations. I have seen no studies linking catchment size and depth of flow-paths to decreased DOC export. That said, the influence of catchment size is generally only a dominant factor for stream DOC characteristics when comparing systems where catchment size or water retention time differ by several order of magnitude (e.g. see Catalan et al 2016, Nature Geoscience). In this case, the difference between a 134 and a 321 km$^2$ catchment may not be enough to see this effect clearly. Also, if difference in catchment size was a dominant control on catchment DOC export, then we would expect the Notawohka catchment to have higher DOC export and higher DOC aromaticity than the Scotty Creek catchment, which is not the case in this study.

A general point here is that we cannot, and do not try to, attribute differences between the Scotty Creek and Notawohka Creek solute export patterns solely to effects of wildfire. That Notawohka was recently affected by wildfire is only one of the several differences between these catchments. That said, we think that the catchment have sufficient similarities in size and land cover composition that we can discuss which differences in solute export patterns between these catchments that are more likely to be due to wildfire. Hence the comparison with the soil pore water, which allows us to say whether differences at the catchment outlets are consistent with observed shifts in soil porewater or peatland hydrology.

**2-This comment is also related to catchment size: is it possible that differences in catchment size influenced the DOC yields (due to microbial respiration during different water transit times)?**

Our response: See above.

**Technical corrections/suggestions:**

**1-Optional idea: Page 8, line 5- "(R2 = 0.71, p < 0.005)." and so forth. -You may add sample size.**

Our response: We have added: *n = 16*

**2-Page 8, line 5- Replace estimated by estimates.**

Our response: Done.

**3-Page 18, line 1- Replace 4.3 by 4.1.**

Our response: Done.

---

## Author Comment (AC2) · 12 Jul 2018

We thank the referee for their comments, which have allowed us to further improve a few key points and overall clarity of the manuscript. Responses to specific comments are attached.

Please also note the supplement to this comment: https://www.hydrol-earth-syst-sci-discuss.net/hess-2018-253/hess-2018-253-AC2-supplement.pdf

253, 2018.

---

## Author Response (AR2)

**Response to Editor, Dr. Sean Carey, regarding comments on manuscript "Seasonal shifts in export of DOC and nutrients from burned and unburned peatland-rich catchments, Northwest Territories, Canada " by Burd et al.,**

Comments to the Author from Editor, Dr. Sean Carey:

*Dear Authors,*

*I would like to thank you for this important contribution to the special issue on understanding and predicting Earth system and hydrological change in cold regions. This paper presents important new insight and information regarding the biogeochemistry of two peatland-dominated systems in Canada's Northwest Territories. The blending of high-frequency absorbance data with flow, solute and 14C data is particularly novel and improves upon our conceptual models in this region. In addition, the limited influence of fire on DOC was also an important observation.*

*In my final review of the manuscript, I have one issue that I believe the authors should reflect on to either temper their statements and/or delete from the manuscript as I do not believe it is of large importance and compared with the other work is overly speculative. Using a relatively weak Q-DOC relationship, historical DOC yield is calculated (1005-2015). On page 14, a value of 2.2+/-0.9 g C m-2 is given and in the discussion (page 21, line 24) this ranges from 0.6-5. While I believe that these are for different periods of the year, my concern is that there is a real possibility (and literature support) to suggest that Q-DOC relationships are not steady and shift with wetness, seasonal delivery of precipitation, etc. The year of the study was comparatively dry in the late season compared to the long-term data which often shows a very wet late summer/fall which would could have large implications on this relationship. The last two sentences in the paragraph beginning on page 21 line 18, particularly the last one regarding the sensitivity of these watersheds compared to other boreal ones, I believe are too strongly stated based on the data here. I would ask the authors to consider my concerns in a final manuscript.*

*Sean Carey, McMaster University*

Our response:

Thanks for the comments. We have made a number of changes to take these comments into account.

We have removed the analysis of historical DOC yields based on the Q-DOC relationship, from the methods section, as well as the two places in the results and discussion as mentioned above in the comment. We have kept a softer statement on the potential influence of climate change on DOC and nutrient yields due to altered runoff generation in the discussion, this section now reads:

**The dry climate of the study region restricted the cumulative catchment DOC yield to < 2 g C m-2 for the study period, which is substantially lower than the range 4 to 15 g C m-2 yr-1 found for boreal catchments in other regions with similar peatland coverage (Lamontagne et al. 2000; Olefeldt et al. 2013b). Runoff during the 2016 study period from the Scotty Creek catchment was 85 mm, below the long term (1995-2015) average of 125 mm for the same period. However, the long term record also**

shows that the region has a very large inter-annual variability in runoff generation, with a range in annual runoff between 30 and 330 mm. This large variability is likely a consequence of the balance between precipitation and evapotranspiration in this dry boreal climate, where even small variability in either precipitation or evapotranspiration causes relatively large variability in runoff. Climate change thus has a large potential to cause altered runoff patterns in the region through altered precipitation or evapotranspiration, which would also strongly influence catchment yields of DOC and nutrients.